# Characterizing Undernourished Children Under-Five Years Old with Diarrhoea in Mozambique: A Hospital Based Cross-Sectional Study, 2015–2019

**DOI:** 10.3390/nu14061164

**Published:** 2022-03-10

**Authors:** Júlia Sambo, Marta Cassocera, Assucênio Chissaque, Adilson Fernando Loforte Bauhofer, Clémentine Roucher, Jorfélia Chilaúle, Idalécia Cossa-Moiane, Esperança L. Guimarães, Lena Manhique-Coutinho, Elda Anapakala, Diocreciano M. Bero, Jerónimo S. Langa, Katja Polman, Luzia Gonçalves, Nilsa de Deus

**Affiliations:** 1Instituto Nacional de Saúde (INS), EN1, Bairro da Vila-Parcela nº 3943, Distrito de Marracuene, Maputo 264, Mozambique; marta.cassocera@ins.gov.mz (M.C.); assucenio.chissaque@ins.gov.mz (A.C.); adilson.bauhofer@ins.gov.mz (A.F.L.B.); jorfelia.chilaule@ins.gov.mz (J.C.); idalecia.moiane@ins.gov.mz (I.C.-M.); esperanca.guimaraes@ins.gov.mz (E.L.G.); lena.manhique@ins.gov.mz (L.M.-C.); elda.anapakala@ins.gov.mz (E.A.); diocreciano.bero@ins.gov.mz (D.M.B.); jeronimo.langa@ins.gov.mz (J.S.L.); nilsa.dedeus@ins.gov.mz (N.d.D.); 2Instituto de Higiene e Medicina Tropical (IHMT), Universidade Nova de Lisboa, 1349-008 Lisboa, Portugal; LuziaG@ihmt.unl.pt; 3Institute of Tropical Medicine (ITM), 2000 Antwerp, Belgium; clementine.roucher@live.fr (C.R.); KPolman@itg.be (K.P.); 4Faculty of Science, Infectious Disease, Vrije Universiteit (VU), 1081HV Amsterdam, The Netherlands; 5Global Health and Tropical Medicine, Instituto de Higiene e Medicina Tropical, Universidade Nova de Lisboa, 1349-008 Lisbon, Portugal; 6Centro de Estatística e Aplicações da Universidade de Lisboa (CEAUL), 1749-016 Lisbon, Portugal

**Keywords:** children, diarrhoea, undernutrition, associated factors, Mozambique

## Abstract

Diarrhoea is associated with undernutrition and this association is related to increased morbidity and mortality in children under-five. In this analysis we aimed to assess the frequency and associated factors of undernutrition in children under-five with diarrhoea. A hospital-based cross-sectional study was conducted from January 2015 to December 2019 through a surveillance system in five sentinel hospitals in Mozambique. Sociodemographic and clinical information was collected, including anthropometry. A total of 963 children were analysed. The overall undernutrition frequency was 54.1% (95% CI: 50.9–57.2), with 32.5% (95% CI: 29.6–35.5) stunting, 26.6% (95% CI: 23.9–29.6) wasting and 24.7% (95% CI: 22.1–27.5) underweight. Children from Nampula province had 4.7 (*p* = 0.016) higher odds for stunting compared with children from Maputo. Children whose caregiver was illiterate had higher odds of being underweight 5.24 (*p* < 0.001), and the wet season was associated with higher odds 1.70 (p = 0.012) of being wasted. Children born under 2500 g of weight had 2.8 (*p* = 0.001), 2.7 (*p* < 0.001) and 2.6 (*p* = 0.010) higher odds for being underweighted, wasted and stunted, respectively. The HIV positive status of the children was associated with higher odds of being underweight 2.6 (*p* = 0.006), and stunted 3.4 (*p* = 0.004). The province, caregiver education level, wet season, child’s birthweight and HIV status were factors associated with undernutrition in children with diarrhoea. These findings emphasise the need for additional caregiver’s education on the child’s nutrition and associated infectious diseases. More studies are needed to better understand the social context in which a child with diarrhoea and undernutrition is inserted.

## 1. Introduction

Diarrhoea is one of the main causes of child morbidity and mortality in developing countries [1] and undernutrition is one of the most important risk factors for this illness globally [2]. In 2019, diarrhoea was the third leading cause of death in children under-five years old with 9.9% of deaths worldwide [3]. In Mozambique, diarrhoea remains a great public health challenge, despite the efforts to reduce its burden in the last two decades [4]. In 2017, it was the fourth major cause of mortality in children under-five years old and responsible for approximately 6.9% of deaths in this age group [4,5]. According to data from a health survey conducted in Mozambique in 2015, the national prevalence was estimated at 11% and higher percentages were observed in children aged 6–11 months and 12–23 months with 18.4% and 19.1%, respectively [6].

Undernutrition in children was shown to increase the risk of occurrence of infectious diseases, death and delayed cognitive development, that can lead to low adult incomes, poor economic growth and an intergenerational transmission of poverty [7]. Globally in 2019, it was estimated that 149 million (21.9%) under-five years old children were stunted and 49.5 million (7.3%) were wasted [8,9]. In 2018, it was estimated that Africa contributed with one third of the stunted cases in children under-five [9]. In Mozambique, in 2015, it was estimated that 42.3% of children under-five years old were stunted, 15.6% were underweighted and 4.4% wasted [10,11,12].

Undernutrition and diarrhoea have a bidirectional association [13]. Diarrhoeal disease significantly affects the nutritional status by interfering with the intestinal absorption of nutrients, and, on the other hand, undernutrition might be a predisposing factor to the onset of diarrhoeal diseases by inducing an alteration of the host’s immunity [13,14]. Both pathologies in the same child significantly increases the risk of a child’s death [7,15,16].

Many socio-economic and clinical factors can contribute to the development of diarrhoeal disease and undernutrition [14,16,17]. In Mozambique, in 2014, 48% of the population was living in poverty [18]. In 2010, UNICEF, reported that 51% of Mozambican children under-five years old with a low height for age (stunting) had diarrhoea in a maximum period of two weeks prior to the data collection [19]. Moreover, studies conducted in the Manhiça district, a rural area of southern Mozambique, reported a high burden of diarrhoea and undernutrition in under-five years old children [20]. Diarrhoea was also identified as one of the risk factors for mortality in children severely undernourished [14].

Several factors have been related to child undernutrition such as the level of education of the caregiver [17]. Moreover, several studies have reported that poor access to clean water and unimproved sanitation are factors associated with undernutrition and diarrhoea, increasing the probability of the child’s death [7,15,16].

Clinical factors such as birth weight are commonly described as being associated with diarrhoeal infection and undernutrition [8,21]. In 2015, 20.5 million (14.6%) children were born with a low birth weight [8]. Besides a low birth weight, parasitic infection is also associated with undernutrition in children [1,22]. Another clinical factor to consider when studying diarrhoea and undernutrition is the Human Immunodeficiency Virus (HIV) infection. HIV is one of the leading health problems in Mozambique [14,20] and in 2009, the prevalence of HIV was 2.3% in children under 12 months of age and 1.7% in children between 1 to 4 years old [23]. A study conducted in 2010 in the Manhiça district, southern Mozambique, showed that stunting (57%), and developmental delay (44%) were high in children under-fifteen years old who were HIV positive, compared to those HIV negative [24]. Another study in children under-five, in the same area, showed that the occurrence of moderate-to-severe diarrhoea among HIV positive patients carried a worse prognosis [14].

In Mozambique, diarrhoea and undernutrition in children are major public health problems [14,20]; however, most of the studies conducted in the country estimated the occurrence of diarrhoea and undernutrition in the south of the country, or only reported an association between undernutrition and diarrhoea caused by a specific pathogen [14,20,22,24,25].

Our study aims to assess the burden of undernutrition in children under-five with diarrhoea in five sentinel sites across the country, representing the three main regions of the country and the potential factors that might be associated with undernutrition conditions.

## 2. Materials and Methods

### 2.1. Study Design and Population

This study uses the data of children under-five enrolled in the National Surveillance of Diarrhoea (ViNaDia). The data was collected from January 2015 to December 2019 in five sentinel sites across the three regions of the country. In the Southern region, Hospital Geral de Mavalane (HGM), Hospital Geral José Macamo (HJM) and Hospital Central de Maputo (HCM) in Maputo city; in the Centre, Hospital Geral de Quelimane (HGQ), in the Zambézia province; and in the North, Hospital Central de Nampula (HCN), in the Nampula province.

Children from 0 to 59 months admitted as inpatients or outpatients with diarrhoea disease were invited to the surveillance through their caregiver. Diarrhoeal disease was defined as the passage of three or more loose or liquid stools in the last 24 h [22,25,26].

### 2.2. Case Report Form

For all eligible children whose legal guardian consented to their inclusion in the surveillance, a case report form (CRF) was filled with sociodemographic and clinical variables. The socio-demographical information of the children was obtained from the child’s caregiver, and the clinical data (malaria, pneumonia, child’s HIV status, and others) was collected from the patient report filled by the clinicians and collected from the child health card.

### 2.3. Anthropometric Measurements

Anthropometric measurements (weight and height/length) were performed on each child by trained nurses. Children’s weight was taken naked or in light clothing. For children under two years of age, weight was measured while the child was lying down or by subtracting the weight of the mother from the weight of the mother with her child when the appropriate scale for babies was not available in the health facility. The length of children under two years or those unable to stand alone, was measured in a recumbent position lying down. The height of children aged two or older was measured while standing.

Undernutrition was defined as (i) stunting (height-for-age (HAZ) z-score < −2), (ii) wasting (weight-for-height (WHZ) z-score < −2), and (iii) underweight (weight-for-age (WAZ) z-score < −2), according to the World Health Organization (WHO) guidelines [27]. Children were considered well-nourished or overweight if their z-scores for WAZ, HAZ, and WHZ were ≥ −2. Children who presented outlier Z-scores were excluded from the analysis [28].

### 2.4. Sample Collection and Laboratory Procedures

A single faecal sample was collected per child for laboratory analysis to test the presence of intestinal parasites. The samples were collected from children to the sample collector receptacle, kept refrigerated in cooler boxes, and then sent to the parasitology reference laboratory at the *Instituto Nacional de Saúde* (INS) in Maputo. Faecal specimens from sentinel sites outside Maputo province were kept at −20 °C until shipment for laboratory analysis.

The presence of parasites was examined by light microscopy using a formol-ether concentration method (protozoans and helminths) and the Modified Ziehl–Neelsen stain was used for opportunistic parasites, namely, *Cryptosporidium spp*., *Cyclospora cayetanensis* and *Cystoisospora belli* [29]. A child was considered positive if at least one parasite was detected by either method. Microscopy results were validated after the slides were read by two independent technicians. In the case of disagreement, an additional reading was made by a third laboratory technician.

### 2.5. Statistical Analysis

The CRF data and the laboratory results were introduced in Epi Info™ V3.5.1 (Centers for Disease Control and Prevention, Atlanta, GA, USA, 2008) by double entry, and exported to IBM SPSS software (Statistical Package for the Social Science, Armonk, NY, USA: IBM Corp, 2011, version 27.0, Chicago, IL, USA). To assess the nutritional status of the children, the Z-scores were calculated using the software, Anthro V.3.2.2, recommended by the WHO [30]. The proportions of stunting, wasting and underweight were estimated through a 95% confidence interval (95% CI) based on the Wilson method, using the EpiTools [31]. Descriptive analyses for qualitative variables were summarized by absolute (n) and relative (%) frequencies. The asymmetric quantitative variables (e.g., the child’s age) were described using median and interquartile intervals. Stunting, wasting and underweight were also used as dependent variables in multiple logistic regression models. The fitted models included explanatory variables (caregiver education level, mother’s marital status, child’s caregiver, caregiver age, type of house, piped water, treated water, season, number of family members, birthweight, someone with diarrhoea at home in prior seven days, malaria, pneumonia, parasitic infection, immunization, and child HIV) with a *p*-value below 0.20 in chi-square/Fisher’s exact tests or other relevant epidemiological variables (child’s age, child’s sex, province, and study year). The Hosmer–Lemeshow test was used for goodness of fit for the logistic regression models. Cox and Snell R square coefficients were calculated to fitted multiple models. Adjusted Odds ratios, with a corresponding 95% CI, were obtained to quantify the magnitude of the associations. A *p*-value < 5% was fixed as the criteria for statistical significance.

## 3. Results

During the study period, 2256 children from 0 to 59 months old with diarrhoea were enrolled, from which 1010 were excluded due to a lack of data to calculate the z-score (sex, weight, or height) to identify their nutritional status. Based on the children’s anthropometric measurements, the Anthro software by the WHO flagged a total of 283 z-score outliers, and they were also excluded from the analysis. The overall final sample size was 963 children (Figure 1). From the 963 children included, 59.6% (574/963) were male, and the median age was 12 months (IQR 9.0–19.0). The overall frequency of undernutrition was 54.1% ((521/963); 95% CI: 50.9–57.2).

Among the 963 children, 32.5% ((313/963); 95% CI: 29.6–35.5) were stunted, 26.6% ((256/963); 95% CI: 23.9–29.6) wasted and 24.7% ((238/963); 95% CI: 22.1–27.5) underweighted (Figure 1). An overlap was observed, with 5.7% ((55/963); 95% CI: 4.4–7.4) between underweight, wasting and stunting, 9.4% ((85/963); 95% CI: 7.8–11.5) between underweight and wasting, and 8.3% ((85/963); 95% CI: 7.1–10.8) between underweight and stunting.

### 3.1. Frequency of Undernutrition Status across Years and Sociodemographic Characteristics

Although no significant differences were found in the proportion of children with undernutrition by year, decreases of the percentage of underweighted [30.0% (36/120) in 2015 to 25.2% (35/139) in 2019], wasted [32.5% (39/120) to 20.9% (29/139)] and stunted [35.0% (42/120) to 32.4% (45/139)] were observed in children between 2015 and 2019, respectively. For stunting, the decrease was 2.6% in this period (Figure 2).

Among the provinces, significant differences were found regarding the percentage of underweight (*p* < 0.001) and stunted (*p* < 0.001) children (Figure 3). Compared with Maputo (20.4%; 127/624), the Zambézia and Nampula provinces presented higher frequencies of underweighted children—35.9% (28/78) and 31.8% (83/261), respectively—and of stunting—47.4% (37/78) and 49.0% (128/261), respectively. Frequencies of wasting were similar among the three provinces (*p* = 0.660) (Figure 3).

Table 1 gives a general description of the sociodemographic characteristics from the children with diarrhoea per type of undernutrition; underweight, wasting, and stunting. Additionally, for the children’s age (in months), a median of 13 months was obtained in all types of undernutrition (IQR 9.0–20.0 for underweight and for stunting), (IQR 8.0–20.0 for wasting).

As presented in Table 1, underweight was more frequent in children whose caregiver was illiterate (48.4%, *p* < 0.001), and in children living in a house made of other types of material than reed, mud and brick (66.7%, *p* < 0.001). Underweight was also more frequent in children living in a household with no piped water (30.7%, *p* < 0.001).

Similarly, wasting was observed more frequently in children whose caregiver was illiterate (38.5%, *p* = 0.019), living in a house made of other types of material (55.6%, *p* = 0.045) different from reed, mud and brick. Wasting was also more frequent during the wet season (31.4%, *p* = 0.003).

Stunting was more frequent in children whose caregiver had no formal education (48.4%, *p* = 0.002), married/co-habitation mothers (35.6%, *p* = 0.005), and children living in a house made by mud (52.1%, *p* < 0.001). Stunting was also more frequent in children without access to piped water (40.8%, *p* < 0.001), and children living in a household with five or more family members (35.5%, *p* = 0.047) (Table 1).

### 3.2. Clinical Characteristics

Table 2 shows that underweight (38.4%, *p* < 0.001) and wasting (39.3%, *p* = 0.001), were more frequent in children born with low weight than children born with normal weight (≥2500 g). Underweight (52.2%, *p* < 0.001), wasting (47.8%, *p* < 0.001) and stunting (55.2%, *p* < 0.001) were more frequent in children HIV positive than in children HIV negative. Underweight was also more frequent in children with an enteric parasitic infection (35.5%, *p* = 0.028).

### 3.3. Factors Associated with Nutritional Status in Children with Diarrhoea

Table 3 shows the selected models for each type of undernutrition. Cox and Snell R square values were 0.146 for underweight, 0.091 for wasting, and 0.151 for stunting.

Children from the Nampula province had 4.7 (OR: 4.68, 95% CI: 1.63–13.41; *p* = 0.016) higher odds for stunting compared to children from Maputo.

Children whose caregiver had no formal education were 5.2 times (OR = 5.16, 95% CI: 2.39–11.13; *p* < 0.001) more likely to become underweighted when compared with those whose caregiver had a secondary or higher education. Children whose caregiver was younger (<21 years) were less likely to be wasted (OR = 0.45, 95% CI: 0.22–0.91; *p* = 0.044).

The wet season was associated with 1.7 (OR = 1.70, 95% CI: 1.12–2.56; *p* = 0.012) higher odds for wasting when compared to the dry season.

Children whose birthweight was less than 2500 g were 2.8 (OR = 2.75, 95% CI: 1.53–4.95; *p* = 0.001), 2.7 (OR = 2.73, 95% CI: 1.57–4.76; *p* < 0.001) and 2.6 (OR = 2.64, 95% CI: 1.26–5.50; *p* = 0.010) times more likely to be underweighted, wasted and stunted, respectively, than children whose birthweight was 2500 g or more.

The odds of being underweighted and stunted were 2.6 (OR = 2.64, 95% CI: 1.33–5.26; *p* = 0.006) and 3.4 (OR = 3.37, 95% CI: 1.46–7.80; *p* = 0.004) times higher for HIV positive children than for those with a negative status for HIV, respectively.

## 4. Discussion

In this analysis, we assessed the frequency and risk factors of undernutrition in children with diarrhoea living in three regions of Mozambique. The overall frequencies of underweight, wasting and stunting were 24.7%, 26.6% and 32.5%, respectively. These values are higher, compared with the prevalence of underweight and wasting in the same age group in other studies in children without diarrhoea, in other settings in Mozambique [10,11,12].

Although the frequency of stunting was lower than the prevalence observed in the same age group in Mozambique at the national level, (43.0% in 2011 and 42.5% in 2015) it was the higher frequency of undernutrition in children with diarrhoea described so far [10,11,12,32]. The results of the present analysis for the three types of undernutrition are aligned with the overall data from the country that shows that stunting is the most frequent type of undernutrition in children under-five [10,11,12,32].

Despite the fact that the prevalence of people living in poverty fell from 59% (2008) to 49% (2014) according to a report on the food crisis in 2019, Mozambique is still one of the 55 countries living with food insecurity [18]. Over the last 20 years, the country has been fustigated by cyclones, floods and droughts that lead to agricultural losses and thus food unavailability, and factors such as extreme weather and conflicts that are known to lead to food insecurity in low and middle income countries [18]. All these factors could help to explain the frequencies of undernutrition observed in the present analysis across the country.

Our results from the Nampula province (49%) for stunting, corroborates with the Demographic Health Survey (DHS) conducted in the country in 2011, where the province presented the highest prevalence (55.3%) of stunted children [32]. Conditions such as food insecurity, high population density, poor health indicators and the highest percentage of poverty (44%) may explain the higher prevalence observed in this province, compared with Maputo and Zambézia [33]. In Nampula, 61.9% of the population consume water from unsafe sources, 46% of the population do not have access to improved latrines and approximately 50% of the population takes 30 to 60 min to access basic services, such as a water source, public transportation, and food markets [34].

Children whose caregiver were illiterate were significantly associated with being underweighted, when compared to those whose caregiver had a secondary or higher education level. Studies conducted in Bangladesh, Nigeria, Kenya, Tanzania, Ethiopia and Mozambique presented the same relation between the mother’s education level and a child’s nutritional status, although those studies were conducted in children without diarrhoea [17,35,36,37,38,39,40]. This association can be explained by the fact that, during formal education, a woman can gain knowledge on health subjects, including nutrition aspects that could help her to better address health issues in her family and community [35,40]. Additionally, formal education could allow a better understanding of how to implement correctly the health advice given by health workers [35,40].

The results of this analysis suggest that children whose caregiver was younger (<21 years) were less likely to be wasted, but this needs to be interpreted carefully as some studies found different associations between undernutrition and a caregiver’s age [36,41]. Another result from the present analysis that must be carefully interpreted is the higher frequency of stunting observed in children whose mother was married/co-habiting, reminding us that in the present multivariate analysis no association was observed between a mother’s marital status and undernutrition. For undernutrition and the mother’s marital status, Fernandes et al. found that a child not living with their father was less likely to be undernourished while Amadu et al. found that children whose mother was married were more likely to be wasted and stunted [41,42]. Both associations, caregiver age and mother’s marital status, should be evaluated using a specific study design that includes the analysis of cofounders to better understand the relationship.

The frequency of undernutrition in children with diarrhoea was higher in children without access to piped water, although no association was observed, but the higher frequency observed can be easily explained by the quality of water from non-piped sources. Non-piped water sources tend to be contaminated with pathogens that can cause diarrhoea. The relation between the quality of water and undernutrition is indirect but well known with poor water quality being associated with stunting and thinness [43].

Drinking unimproved water was not significantly associated with any type of undernutrition. This could be explained by the fact that in some families even with treated water available, the members of the family would rather drink the untreated water because some treatment methods can change the taste of the water [44]. Another plausible justification would be the incorrect implementation of known methods to treat drinking water [16]. The results of the present analysis were similar to those found in studies conducted in Ghana and Tigray [16,45]. On the contrary, a study conducted in children aged 5–24 months in Cameroon showed that unimproved drinking water was associated with underweight, wasting and stunting [36]. Another study also revealed an association with stunting in the Democratic Republic of Congo [46]. The higher risk of undernutrition due to drinking unsafe water is related to water borne diseases by exposure to enteric pathogens [47].

A significative difference was observed between the wet and dry season with wasting. Children aged under-five with diarrhoea had higher odds of being wasted in wet compared with dry seasons. During the wet season, diarrhoea diseases are more frequently observed which is already known as a risk factor for undernutrition [16] due to food insecurity, food access and availability [48].

The number of family members has been reported as a risk factor for the occurrence of undernutrition in children under-five with diarrhoea [16]. Although in the present analysis no association was observed, the association between these two factors is justified through aspects related to hygiene, sanitation, family income, food access and availability [49].

Although the results of this study were not statistically significant for the association between malaria and undernutrition in children under-five, malaria is commonly reported to be a predictor for the occurrence of undernutrition [50].

There was a positive association between all type of undernutrition and low birthweight in the present analysis. A child born with a low weight has indeed a higher risk of developing wasting, underweight and stunting, than a child born with a normal weight [21,51]. For example, using Malawian DHS data, it was shown that a child born with a low weight is at risk of remaining undernourished during the first years of their life [51]. Additionally, low birthweight is described as a factor associated with an increased vulnerability to diarrhoeal infection [21].

In the public health literature, HIV is one of the factors identified as being associated with undernutrition [14,52,53]. Thus, many studies have showed that the frequency of underweight, wasting and stunting is higher in HIV positive children compared to HIV negative children. HIV positive children have a higher risk of developing undernutrition, and especially, underweight and stunting [52,53]. Studies have showed that the association between undernutrition and HIV in children under-five years of age, results from a reduction of food intake and poor absorption of nutrients, that is common in children HIV positive with diarrhoea [20,52,53].

Since the data analysed were from a hospital-based surveillance, and the health workers collected all the data during the routine treatments, there are two identified limitations. The first limitation is the missing information related to the independent and dependent variables that reduced the sample size, from an initial 2256 to 963 for the analysis. The second limitation is inherent to cross-sectional studies that does not allow to establish a direct cause–effect relation between the collected variables and undernutrition in these children with diarrhoeal disease. Despite the limitations, due to the lack of data on the subject in our country, this analysis brings important data on factors associated with undernutrition in children with diarrhoea that can guide future studies, design actions for prevention and can improve childcare.

## 5. Conclusions

The frequency of stunting varied across the provinces, being higher in the Nampula province. The caregiver education level (illiterate) was associated with being underweight. The wet season was associated with being wasted. Children born with a low birthweight were related to the three types of undernutrition, and a HIV positive status was related to underweight and stunting.

The findings of this analysis emphasise the need for additional mother’s education activities on children’s health with a focus on diarrhoea and undernutrition and infectious diseases frequently associated with diarrhoea. Our results highlight the need to address the efforts during the antenatal care and children’s health visits to disseminate and promote actions that can help prevent the development of any type of undernutrition. The findings of this analysis also highlight the importance of the integrated management of childhood illness, which can help reduce the morbi-mortality in children due to health conditions associated with undernutrition.

Studies to understand better the social context of children with diarrhoea and undernutrition are also recommended to evaluate the strength and weakness of the health system to implement an integrated management of diseases in children.

## Figures and Tables

**Figure 1 nutrients-14-01164-f001:**
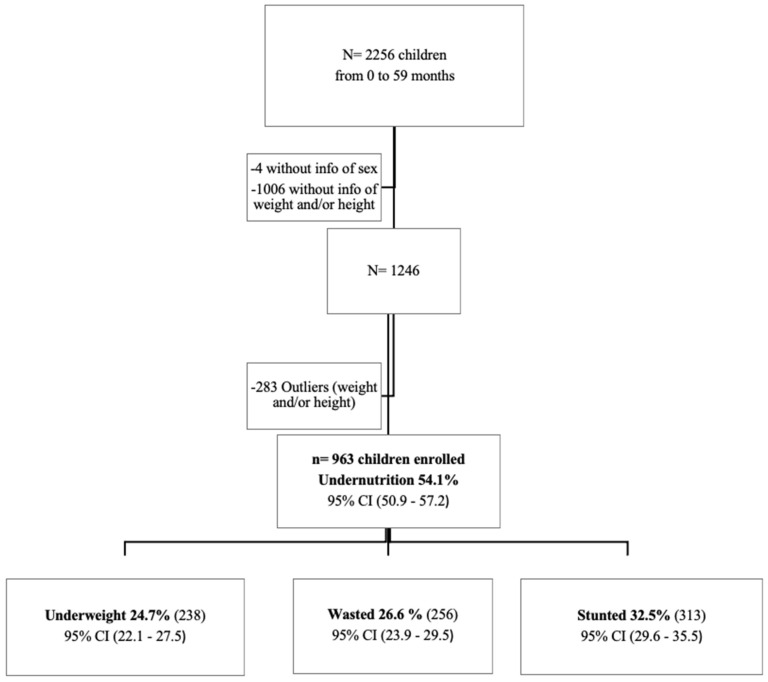
Flow diagram of children enrolled for the analysis and their nutritional status.

**Figure 2 nutrients-14-01164-f002:**
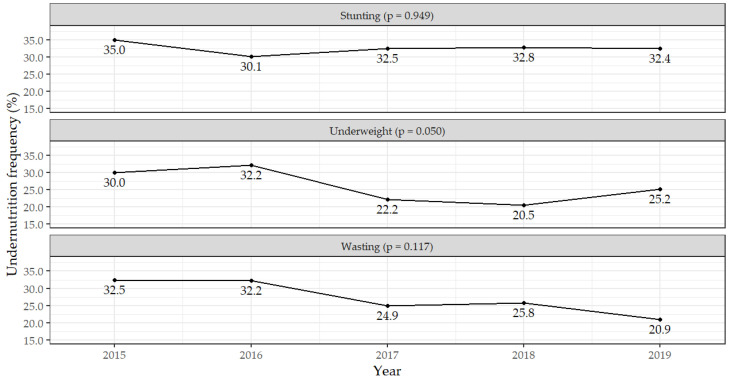
Frequency of undernutrition type in children by year of admission.

**Figure 3 nutrients-14-01164-f003:**
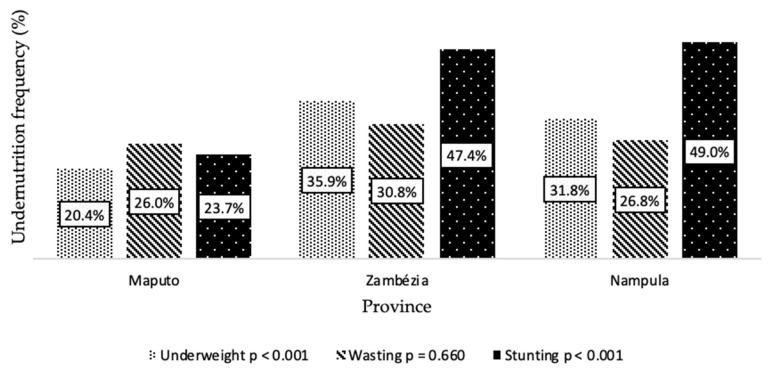
Frequency of different undernutrition status in children by provinces.

**Table 1 nutrients-14-01164-t001:** Sociodemographic characteristics of the study children for the total sample and frequencies of underweight, wasting, and stunting for each category.

Characteristic	Total		Underweight		*p*-Value	Wasting		*p*-Value	Stunting		*p*-Value
	n	(%)	n	(%)		n	(%)		n	(%)	
Total	963	100	238	24.7		256	26.6		313	32.5	
**Sex**					0.350 ^a^			0.813 ^a^			0.733 ^a^
Male	574	59.6	148	25.8		151	26.3		189	32.9	
Female	389	40.4	90	23.1		105	27.0		124	31.9	

**Child age group (in months)**					0.398 ^a^			0.866 ^a^			0.080 ^a^
0–5	65	6.7	13	20.0		20	30.8		12	18.5	
6–11	365	37.9	84	23.0		94	25.8		120	32.9	
12–23	374	38.8	95	25.4		99	26.5		124	33.2	
24–59	159	16.5	46	28.9		43	27.0		57	35.8	
**Caregiver Education Level**					**<0.001 ^a^**			**0.019 ^a^**			**0.002 ^a^**
No education	91	9.4	44	48.4		35	38.5		44	48.4	
Primary	369	38.3	95	25.7		99	26.8		122	33.1	
Secondary/higher	495	51.4	98	19.8		120	24.2		146	29.5	
Missing	8	0.8	-	-		-	-		-	-	
**Mother’s marital status**					0.544 ^b^			0.788 ^b^			**0.005 ^b^**
Single	287	29.8	65	22.6		80	27.9		72	25.1	
Married/co-habitation	637	66.1	163	25.6		167	26.2		227	35.6	
Divorced/widower	13	1.3	4	30.8		4	30.8		4	30.8	
Missing	26	2.7	-	-		-	-		-	-	
**Child’s caregiver**					0.333 ^a^			0.567 ^a^			0.213 ^a^
Mother	870	90.3	220	25.3		235	27.0		289	33.2	
Others	83	8.6	17	20.5		20	24.1		22	26.5	
Missing	10	1.0	-	-		-	-		-	-	
**Caregiver age (in years)**					0.319 ^a^			0.129 ^a^			0.253 ^a^
<21	162	16.8	43	26.5		43	26.5		61	37.7	
21–30	537	55.8	121	22.5		130	24.2		166	30.9	
>30	241	25.0	65	27.0		75	31.1		75	31.1	
Missing	23	2.4	-	-		-	-		-	-	
**Type of house**					**<0.001 ^b^**			**0.045 ^b^**			**<0.001 ^b^**
Reed	45	4.7	20	44.4		16	35.6		16	35.6	
Mud	219	22.7	77	35.2		65	29.7		114	52.1	
Brick	674	70.0	132	19.6		167	24.8		175	26.0	
Others	9	0.9	6	66.7		5	55.6		3	33.3	
Missing	16	1.7	-	-		-	-		-	-	
**Piped water**					**<0.001 ^a^**			0.101 ^a^			**<0.001 ^a^**
Yes	540	56.1	106	19.6		131	24.3		141	26.2	
No	407	42.3	125	30.7		119	29.1		167	40.8	
Missing	16	1.7	-	-		-	-		-	-	
**Treated water**					0.452 ^a^			0.094 ^a^			0.145 ^a^
Yes	349	36.2	89	25.5		80	22.9		122	35.0	
No	503	52.5	117	23.3		141	28.0		152	30.2	
Missing	111	11.5	-	-		-	-		-	-	
**Season**					0.066 ^a^			**0.003 ^a^**			0.911 ^a^
Wet	424	44.0	117	27.6		133	31.4		137	32.3	
Dry	539	56.0	121	22.4		123	22.8		176	32.7	
**Family members in household**					0.794 ^a^			0.563 ^a^			**0.047 ^a^**
<5	435	45.2	110	25.3		115	26.4		127	29.2	
≥5	422	43.8	110	26.1		119	28.2		150	35.5	
Missing	106	11.0	-	-		-	-		-	-	

^a^: Chi-square test; ^b^: Fisher’s exact test; Bold: Significant *p*-values.

**Table 2 nutrients-14-01164-t002:** Clinical characteristics of children and frequency of undernutrition status by category.

Characteristic	Total		Underweight		*p*-Value	Wasting		*p*-Value	Stunting		*p*-Value
	N	(%)	N	(%)		N	(%)		N	(%)	
Total	963	100	238	24.7		256	26.6		313	32.5	
**Birthweight**					**<0.001 ^a^**			**0.001 ^a^**			0.116 ^a^
<2500 g	112	11.6	43	38.4		44	39.3		43	38.4	
≥2500 g	749	77.8	165	22.0		179	23.9		232	31.0	
Missing	102	10.6	-	-		-	-		-	-	
**Someone with diarrhoea at home (Last 7 days)**					0.255 ^a^			0.582 ^a^			0.603 ^a^
Yes	113	11.7	33	29.2		33	29.2		38	33.6	
No	763	79.2	185	24.2		204	26.7		238	31.2	
Missing	87	9.0	-	-		-	-		-	-	
**Malaria**					0.080 ^a^			0.159 ^a^			0.544 ^a^
Yes	58	6.0	20	34.5		20	34.5		21	36.2	
No	887	92.1	215	24.2		231	26.0		287	32.4	
Missing	18	1.9	-	-		-	-		-	-	
**Pneumonia**					0.784 ^b^			1.000 ^b^			0.343 ^a^
Yes	18	1.9	5	27.8		5	27.8		4	22.2	
No	930	96.6	230	24.7		247	26.6		305	32.8	
Missing	15	1.6	-	-		-	-		-	-	
**Parasitic Infection**					**0.028 ^a^**			0.254 ^a^			0.194 ^a^
Positive	93	9.7	33	35.5		30	32.3		37	39.8	
Negative	661	68.6	164	24.8		176	26.6		218	33.0	
Missing *	209	21.7	-	-		-	-		-	-	
**Immunization**					0.226 ^a^			0.548 ^a^			0.615 ^a^
Yes	801	83.2	193	24.1		211	26.3		259	32.3	
No	157	16.3	45	28.7		45	28.7		54	34.4	
Missing	5	0.5	-	-		-	-		-	-	
**Child HIV status**					**<0.001 ^a^**			**<0.001 ^a^**			**<0.001 ^a^**
Positive	67	7.0	35	52.2		32	47.8		37	55.2	
Negative	630	65.4	127	20.2		148	23.5		166	26.3	
Missing	266	27.6	-	-		-	-		-	-	

* Sample was not analysed due to quality issue, or no sample was collected. ^a^: Chi-square test; ^b^: Fisher’s exact test; Bold: Significant *p*-values.

**Table 3 nutrients-14-01164-t003:** Adjusted odds ratio and corresponding confidence interval obtained through multiple logistic regression models for each undernutrition status of children.

Characteristic	Underweight		Wasting		Stunting	
	OR (95% IC)	*p*-Value	OR (95% IC)	*p*-Value	OR (95% IC)	*p*-Value
**Year**		0.100		0.223		0.126
2015	4.71 (1.40–15.83)	**0.012**	3.30 (0.97–11.22)	0.560	4.76 (1.11–20.42)	**0.036**
2016	1.32 (0.63–2.79)	0.466	1.55 (0.75–3.22)	0.238	0.68 (0.28–1.61)	0.380
2017	0.95 (0.50–1.80)	0.882	1.08 (0.59–1.99)	0.802	1.40 (0.71–2.79)	0.332
2018	0.96 (0.49–1.87)	0.906	1.49 (0.80–2.78)	0.205	1.40 (0.56–3.48)	0.457
2019	Ref.		Ref.		Ref.	
**Province**		0.127		0.919		0.016
Maputo	Ref.		Ref.		Ref.	
Zambézia	2.65 (0.99–7.10)	0.052	1.21 (0.45–3.23)	0.704	2.99 (0.82–10.87)	0.097
Nampula	2.00 (0.82–4.88)	0.128	1.03 (0.43–2.45)	0.954	4.68 (1.63–13.41)	**0.004**
**Sex**		0.749		0.424		0.437
Male	1.08 (0.68–1.70)		1.19 (0.78–1.81)		0.82 (0.49–1.36)	
Female	Ref.		Ref.		Ref.	
**Child age group (in months)**		0.052		0.882		0.253
0–5	Ref.		Ref.		Ref.	
6–11	2.19 (0.69–6.94)	0.184	0.81 (0.35–1.90)	0.630	3.22 (0.95–10.98)	0.061
12–23	2.64 (0.84–8.23)	0.095	0.92 (0.40–2.10)	0.835	3.54 (1.03–12.15)	**0.044**
24–59	4.36 (1.31–14.47)	**0.016**	1.08 (0.43–2.67)	0.874	3.04 (0.81–11.39)	0.099
**Caregiver Education Level**		**<0.001**		0.079		0.141
No education	5.16 (2.39–11.13)	**<0.001**	2.06 (0.97–4.36)	0.060	1.41 (0.56–3.57)	0.467
Primary	1.39 (0.85–2.28)	0.195	0.88 (0.56–1.39)	0.585	1.73 (1.00–2.97)	**0.049**
Secondary/above	Ref.		Ref.		Ref.	
**Mother’s marital status**						0.792
Single					Ref.	
Married/co-habitation					0.84 (0.14–5.01)	0.846
Divorced/widower					0.69 (0.12–3.99)	0.680
**Care giver age (in years)**		0.327		**0.044**		
<21	0.89 (0.44–1.82)	0.757	0.45 (0.22–0.91)	**0.026**		
21–30	0.68 (0.26–1.72)	0.150	0.62 (0.39–0.99)	**0.046**		
>30	Ref.		Ref.			
**Type of house**		0.685		0.888		0.611
Reed	Ref.		Ref.		Ref.	
Mud	0.50 (0.16–1.53)	0.224	1.08 (0.34–3.42)	0.900	1.30 (0.35–4.85)	0.697
Brick	0.67 (0.26–1.72)	0.406	1.34 (0.50–3.60)	0.561	0.70 (0.22–2.29)	0.559
Others	-	-	2.19 (0.10–46.32)	0.614	0.55 (0.03–9.35)	0.682
**Piped water**		0.931		0.054		
Yes	Ref.		Ref.		Ref.	0.526
No	1.03 (0.58–1.82)		1.66 (0.99–2.79)		0.81 (0.41–1.57)	
**Treated water**		1.000		0.125		0.774
Yes	1.00 (0.62–1.62)		0.70 (0.45–1.10)		1.08 (0.63–1.85)	
No	Ref.		Ref.		Ref.	
**Season**		0.269		**0.012**		
Wet	1.29 (0.82–2.02)		1.70 (1.12–2.56)			
Dry	Ref.		Ref.			
**Family members in household**						0.178
<5					Ref.	
≥5					0.675 (0.38–1.20)	
**Birthweight**		**<0.001**		**<0.001**		**0.010**
<2500 g	2.75 (1.53–4.95)		2.73 (1.57–4.77)		2.64 (1.26–5.50)	
≥2500 g	Ref.		Ref.		Ref.	
**Malaria**		0.415		0.415		
Yes	0.61 (0.18–2.10)		1.55 (0.54–4.46)			
No	Ref.		Ref.			
**Parasitic Infection**						0.882
Positive					1.06 (0.49–2.32)	
Negative					Ref.	
**Child HIV status**		**0.006**		0.061		**0.004**
Positive	2.64 (1.33–5.26)		1.88 (0.97–3.63)		3.37 (1.46–7.80)	
Negative	Ref.		Ref.		Ref.	

The blank cells in the table mean that the variable did not satisfy the entry criteria to the final models; Bold: Significant *p*-values.

## Data Availability

The data from this study is not publicly available due to restrictions present in the consent forms. The data presented in this study are available on request from the corresponding author.

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
