# Peer review of "Characterizing Undernourished Children Under-Five Years Old with Diarrhoea in Mozambique: A Hospital Based Cross-Sectional Study, 2015–2019"

_nutrients, 2022, doi:10.3390/nu14061164_

Round 1

Reviewer 1 Report

Thank you for submitting this paper investigating factors associated with undernutrition in children under five years suffering from diarrhoea in Mozambique. As you point out this is a cross sectional study and therefore the factors can only be viewed as an association and not directly causal.  However, this is an important subject and one which should be afforded due weight.

On line 121, you use the term "scaling", do you mean that the children were weighed using scales ?

On line 138,  you use the term "recipient" and I think you mean receptacle.

In figure 1 236 outliers were excluded, it would be helpful for the reader if you could add a sentence into the results about the criteria you used for excluding them. 

Author Response

Thank you for submitting this paper investigating factors associated with undernutrition in children under five years suffering from diarrhoea in Mozambique. As you point out this is a cross sectional study and therefore the factors can only be viewed as an association and not directly causal.  However, this is an important subject and one which should be afforded due weight.

On line 121, you use the term "scaling", do you mean that the children were weighed using scales?

Response: Thank you for this important observation, yes, this is what we wanted to say. We revised the sentence. Line 121.

On line 138, you use the term "recipient" and I think you mean receptacle.

Response: Yes, we mean receptacle. We have edited as suggested, thank you. Line 137.

In figure 1 236 outliers were excluded, it would be helpful for the reader if you could add a sentence into the results about the criteria you used for excluding them. 

Response: Thank you for the suggestion. We first mention the exclusion of the outliers in the material and methods section (2.3 Anthropometric measurements) line 131 and we apologize for not being clear. We added a sentence in the results section as suggested. Line 176 to 178.

Reviewer 2 Report

According to the WHO data, diarrhoeal disease is the second leading cause of death in children under five years old, and is responsible for killing around 525 000 children every year. Children who are malnourished or have impaired immunity as well as people living with HIV are most at risk of life-threatening diarrhoea. Authors presented additional and up-to-date evidence regarding the problem.

The manuscript is well structured and methodologically sound. All necessary details are covered in Materials and Methods and Result sections. The discussion is appropriate, including the limitations.

Author Response

According to the WHO data, diarrhoeal disease is the second leading cause of death in children under five years old, and is responsible for killing around 525 000 children every year. Children who are malnourished or have impaired immunity as well as people living with HIV are most at risk of life-threatening diarrhoea. Authors presented additional and up-to-date evidence regarding the problem.

The manuscript is well structured and methodologically sound. All necessary details are covered in Materials and Methods and Result sections. The discussion is appropriate, including the limitations.

Response: We would like to thank the reviewer for the comment to the manuscript.
